# Recent Advances in Electrosynthesized Molecularly Imprinted Polymer Sensing Platforms for Bioanalyte Detection

**DOI:** 10.3390/s19051204

**Published:** 2019-03-09

**Authors:** Robert D. Crapnell, Alexander Hudson, Christopher W. Foster, Kasper Eersels, Bart van Grinsven, Thomas J. Cleij, Craig E. Banks, Marloes Peeters

**Affiliations:** 1Faculty of Science & Engineering, Div. of Chemistry & Environmental Science, Manchester Metropolitan University, John Dalton Building, Chester Street, Manchester M1 5GD, UK; r.crapnell@mmu.ac.uk (R.D.C.); a.hudson@mmu.ac.uk (A.H.); Chris.W.Foster@mmu.ac.uk (C.W.F.); c.banks@mmu.ac.uk (C.E.B.); 2Sensor Engineering, Faculty of Science and Engineering, Maastricht University, P.O. Box 616, 6200 MD Maastricht, The Netherlands; kasper.eersels@maastrichtuniversity.nl (K.E.); bart.vangrinsven@maastrichtuniversity.nl (B.v.G.); thomas.cleij@maastrichtuniversity.nl (T.J.C.); 3School of Engineering, Newcastle University, Newcastle Upon Tyne NE1 7RU, UK

**Keywords:** electropolymerization, molecularly imprinted polymers (MIPs), electrosynthesis, sensors, biomolecules, bacteria, proteins

## Abstract

The accurate detection of biological materials has remained at the forefront of scientific research for decades. This includes the detection of molecules, proteins, and bacteria. Biomimetic sensors look to replicate the sensitive and selective mechanisms that are found in biological systems and incorporate these properties into functional sensing platforms. Molecularly imprinted polymers (MIPs) are synthetic receptors that can form high affinity binding sites complementary to the specific analyte of interest. They utilise the shape, size, and functionality to produce sensitive and selective recognition of target analytes. One route of synthesizing MIPs is through electropolymerization, utilising predominantly constant potential methods or cyclic voltammetry. This methodology allows for the formation of a polymer directly onto the surface of a transducer. The thickness, morphology, and topography of the films can be manipulated specifically for each template. Recently, numerous reviews have been published in the production and sensing applications of MIPs; however, there are few reports on the use of electrosynthesized MIPs (eMIPs). The number of publications and citations utilising eMIPs is increasing each year, with a review produced on the topic in 2012. This review will primarily focus on advancements from 2012 in the use of eMIPs in sensing platforms for the detection of biologically relevant materials, including the development of increased polymer layer dimensions for whole bacteria detection and the use of mixed monomer compositions to increase selectivity toward analytes.

## 1. Scope of Review

Molecularly imprinted polymers (MIPs) have been utilised in sensing platforms as biomimetic recognition elements for a wide selection of targets, ranging from small molecules to larger macromolecules, including cells and bacteria. Extensive reviews have been published in the field of MIP based sensing [1,2,3,4,5]. Electrochemical detection methods, such as electrochemical impedance spectroscopy (EIS), square-wave voltammetry (SWV), and differential pulse voltammetry (DPV), are some of the most popular methods to use in conjunction with MIPs. Even so, the amount of these sensing platforms that utilise electropolymerization to form the MIPs is low. In 2004, it was reported that only around 18% of all electrochemical MIP sensors were produced by electropolymerization, whereas the majority were obtained via standard free radical polymerization [6]. Electropolymerization has numerous advantages over traditional methods of preparation, such as superior adherence to the transducer surface, speed of preparation, the possibility of aqueous preparation, as well as control of the layer thickness and morphology. However, there are areas in which advancements in electropolymerization need to be made. These include the removal of imprinted templates and the inability to simultaneously optimize the conditions for both imprinting and re-binding, such as the solubility and minimization of interactions between the solvent, target, and monomer. Sharma et al. [7] published a review on the use of electrochemically synthesized MIPs (eMIPs) in chemical sensing, which focused predominantly on the detection of small molecules. Since then, research into eMIPs for the detection of small molecules has progressed and significant steps into the detection of larger bio-macromolecules have been made. Hence, this review will critically analyze the production and implementation of electrosynthesized MIPs (eMIPs) into sensing platforms for the detection of biologically important targets.

### 1.1. Molecularly Imprinted Polymers

Molecularly imprinted polymers (MIPs) are functional porous materials with high affinity binding sites that are complementary in size, dimension, and functionality to the analyte of interest [8,9]. Molecular imprinting technology was first developed by Polyakov [10] in the 1930s, with further progression by Wulff [11] in the 1970s, but increased in popularity when Mosbach and co-workers [12] introduced the non-covalent imprinting approach in the 1990s. The non-covalent approach allows for a more flexible choice of monomers and facilitates easy removal of the template from the polymer matrix. The latter can be challenging since template leaching is considered a common problem for the use of MIPs, both covalent and non-covalent [13]. Contrary to antibodies, their natural counterparts, MIPs are low-cost, offer straightforward production [14], are thermally and chemically stable [15], are not sourced from animals, are re-useable, and can be stored at room temperature for long periods. Drawbacks that need to be considered include the time required for development and optimization, which can be overcome using high-through screening or computational methods, and adapting the polymers to work in aqueous environments since water interferes with non-covalent interactions between monomer and template [6]. MIPs have been developed for small ions (known as ion imprinted polymers) ranging to large macromolecules, and this versatility means they can be employed for numerous applications, including solid-phase separation [16], purification [17], catalysis [18], sensors [3], drug delivery [19], and therapeutic applications [20]. The main commercial application lies in the removal or extraction of low-level contaminants since MIPs are powerful tools for the selective extraction of compounds in complex matrices [21] (e.g., whole blood, plasma, urine, food samples, environmental samples, etc.). First, we will give a brief overview of the different methodologies used to produce MIPs and how they are integrated into sensing platforms.

The traditional way of producing MIPs uses free radical polymerization where a rigid monolith is obtained, which is then ground and sieved to obtain particles. This is a simple and scalable approach, but its disadvantages include its inefficiency, substantial loss of polymer in the production process, slow mass transfer, and heterogeneity of the obtained particles [22,23]. Despite these apparent drawbacks, free radical polymerization is still the most common route for MIP production. To integrate these particles into sensors, an adhesive layer can be used into which particles can be impregnated. There have been reports in the literature of direct mixing of particles with screen-printing ink [24,25,26], which significantly speeds up the functionalization process, but leads to limited surface coverage.

To reduce the particle size and prepare homogenous nanobeads, it is possible to use precipitation polymerization [27]. However, the disadvantage is that larger volumes of solvent are required, which interferes with the greenness of the reaction and larger amounts of the imprint molecule are needed. There are reports in the literature, such as by Ye et al., demonstrating that beads obtained by precipitation polymerization outperform particles obtained by grinding and sieving in certain binding assays [28]. 

Emulsion core-shell polymerization can be performed in water-based systems and has the ability to fine-tune the particle size between small nanobeads going to bigger nanoparticles [29]. This procedure requires more optimization compared to the standard bulk polymerization and the polymerization in water can reduce the binding affinity to the target. Significant advantages of this method include control over the size, homogeneity, increased surface area, and the possibility of scaling this up to large reactors. The production of colloidal particles with mini-emulsion was reported by Landfester et al., who used the bifunctional crosslinker, N,O-bismehtacryloylethanolamine (NOBE), for synthesis. As a case study, it was shown that these MIPs significantly increased the imprint factor for the detection of testosterone [30]. However, NOBE is not commercially available and will need to be synthesized first, whereas for free radical polymerization, usually standard commercially available monomers and cross-linkers are used.

The groups of Piletsky and Haupt have described approaches for the solid-phase synthesis of MIP nanoparticles (nanoMIPs, Figure 1 [31]). The first approach consists of the immobilization of the template or a similar molecule onto glass beads around which the polymer is formed. Unreacted monomers and low-affinity monomers were removed at low temperatures, while the high affinity particles were collected at elevated temperatures [32]. This leads to uniform particles with affinity constants that are similar to antibodies, enabling the direct replacement of antibodies with MIPs in enzyme-linked immunosorbent assays (ELISA) [33]. This approach has been developed for a range of targets and has the advantage that the template can be recycled in the reactor, which is particularly important when working with expensive proteins [34]. 

The approach by Haupt focuses on the attachment of an affinity ligand of the protein to the solid support, which enables direct immobilization of the protein [35]. After polymerization around the template, the polymers are separated based on their thermoresponsive properties. These reported nanoparticles have the potential to be biocompatible, enabling them for use in in-vivo sensing [31]. 

These methodologies focus on the formation of the MIP, which then need to be attached to the surface of a transducer. Instead of grafting the template molecule to the surface, it is possible to attach an initiator on an electrode surface and form polymer brushes. This can be combined with a range of techniques that are able to control the polymerization, including, but not limited to, atom transfer radical polymerization (ATRP) [36], reversible addition fragmentation chain-transfer polymerization RAFT [37], nitroxide mediated polymerization (NMP) [33], and iniferter methods [34]. The direct advantages of these methods are faster mass-transfer and better control over the surface morphology. However, it is required to first attach the initiator to the surface and this involves an additional step in the reaction. Common routes to achieving this include the use of self-assembled monolayers for gold electrodes, while for other metals, silanization is often used [38]. The amount of imprint sites is dictated by the amount of initiator on the surface, which is often difficult to determine. This facile approach is not scalable, but leads to a high affinity and can be used to integrate sensors into biomedical devices [39]. 

Soft lithography has often been used for surface imprinting and can be achieved without the need of sophisticated equipment [40]. A pre-polymerized layer is applied onto a transducer surface and after stamping the template into the layer, the polymer layer is fully cured. Dickert et al. pioneered this approach with polyurethane layers, and demonstrated that the amount and distribution of the template on the stamp is crucial to tailor the sensor response [41,42,43,44]. 

Microcontact imprinting involves bringing the electrode, containing a pre-polymerization mixture, in contact with a cover slip and exposing it to UV irradiation. The polymer mixture should have a higher affinity for the electrode structure, while non-reacted monomer and unbound template can be washed away from the mixture [45]. It enables rapid and parallel synthesis of MIPs, and only requires a limited amount of template since a few microliters of solution will be sufficient. While this technique has been developed on silicon wafers and on glass slides, it is less straightforward on other electrode surfaces [46]. It also should be considered that it might be complicated on wires or other electrode configurations, which could limit its application for the use of biomedical devices. 

There are currently over 1000 patents on molecular imprinting, with a range of commercially available products available for filtration and purification. They have been utilized in chromatography columns due to the simple packing process of the particles. Whereas this is a scalable approach, there is no process in place that can mass-produce MIPs with high affinity into electrodes. Peeters et al. [24] described a method for the mass-production of MIPs onto screen-printed carbon electrodes (SPCEs) by the direct mixing of particles with the ink. However, these microparticles have limited affinity and only ~30% of the particles can be taken up in the ink. This may require adaptation of the ink to reflect the nature of the MIP particles; for hydrophilic particles, the ink will need to be altered [47]. The use of nanoparticles as developed by technology in Leicester [32,34] opens up new functionalization procedures since they are not cross-linked and are therefore water-soluble. They also produce a superior affinity since they are pre-selected, through temperature elution, and can be produced in larger volumes. At the present time, MIPs are an emerging technology and cannot compete with traditional antibodies. In the future, it is expected that this approach is more suitable for sensor applications and have the potential to replace antibodies in standard assays [33,48,49]. Molecular imprinting is an emerging technology and to produce polymers with a similar affinity to antibodies, more optimization is required per target. Some examples of commercial outputs utilizing MIPs include SupelMIP (solid phase extraction (SPE) cartridges for drugs and pesticides), AFFINIMIP^®^ SPE (SPE for contaminants and endocrine disruptors), Biotage (industrial resins to remove trace pesticides), Ligar (recovering target molecules from plants, isolating flavor molecules, and extracting contaminants from liquids), and MIPDiagnostics (bespoke nanoparticle synthesis, which can be used in various applications, including sensing platforms).

### 1.2. Electrosynthesis of Molecularly Imprinted Polymers

Generally, the electrosynthesis of molecularly imprinted polymers (eMIPs) takes place via a process called electropolymerization. Electropolymerization is a deposition procedure in which a conductive polymer layer is formed or coated upon an electrode/supporting substrate material in the presence of the desired template. Such electrochemical coatings are carried out utilising a typical three-electrode setup, where a working electrode (i.e., where the coating occurs), reference electrode (typically Ag/AgCl or saturated calomel electrode, SCE), and a counter electrode (usually platinum or nickel) are used within a solution containing a monomer, solvent, and a supporting electrolyte; all of which are vital to the surface morphology of the polymeric film [50]. 

Electropolymerization can be achieved through a variety of electrochemical techniques, namely, voltammetric [51], potentiostatic [52], and galvanostatic [53]. Voltammetric electropolymerization is by far the most popular fabrication route for the electrosynthesis of polymeric layers. In this technique, the sweeping of a set range of potentials, via cyclic voltammetry, between the limits of the monomer oxidation and the reduction of the polymerized polymer occurs. Upon control of the voltage range, the polymer film can be fabricated with varying thicknesses and polymer oxidation. This technique also allows the user to control the speed/sweep rate at which the monomer reacts at the electrode surface. Potentiostatic electropolymerization occurs via the application of a constant potential; the identification of this potential (normally after consultation of the voltammetric behaviour) is vital to the stability, conductivity, and thickness of the polymer layer upon the electrode surface. Galvanostatic electropolymerization is similar to that of the potentiostatic route; however, this involves the application of a constant current to induce polymerization [7,50,54]. 

One indirect sensing mechanism of eMIPs that is frequently used occurs through the same steps as seen in Figure 2. The chosen monomer and desired template are mixed in solution at a specified ratio. This mixture is then left to allow the specific interactions, such as hydrogen bonds and electrostatic interactions, to form. Electropolymerization is then performed, forming a layer comprised of the polymer with the trapped target on the surface of the transducer. The template is subsequently removed from the polymer layer, which is typically one of the more challenging steps of the process. When the target is re-introduced into the system, it binds to the cavities present in the polymer layer, producing a measurable change in either the electrochemical or thermal properties.

Typically, within the academic literature, electroactive monomers, such as o-phenylenediamine and aniline, are regularly utilised as electropolymerizable monomers for the creation of eMIPs. One of the most common polymers used in electrochemical MIP synthesis is polypyrrole (PPy). Since the first research article using a PPy film as an anion sieve in 1986 [55], there have been over 160 publications utilising PPy as an eMIP. The focus on this polymer primarily stems from its ease of oxidation and water solubility. PPy possesses several advantageous intrinsic properties, such as good conductivity, environmental stability, and good redox properties [56]. When subjected to large positive potentials, PPy can be overoxidized, leading to the incorporation of carbonyl groups into the backbone of the polymer, an increase in film thickness, and loss in conductivity [57,58,59]. These phenomena will occur at lower potentials when in an oxygen or water containing environment [60]. Although the loss in conductivity is a disadvantage, a wide range of bioanalytical applications have resulted from imprinting overoxidized PPy layers [61,62,63], which can be incorporated into sensing platforms.

As mentioned previously, Py, is not the only monomer for the creation of eMIPs. Presented within Table 1 are a summary of the common monomers utilised throughout the literature. 

## 2. Biosensing Platforms Incorporating Electrosynthesized Molecularly Imprinted Polymers

### 2.1. Neurotransmitters

Neurotransmitters are chemicals found in the body, responsible for the transmission of signals between neurons or non-neuronal body cells across chemical synapses. There have been multiple reviews on the detection of neurotransmitters. These can cover the detection of a wide range of neurotransmitters [82,83,84] or focus on a specific neurotransmitter, such as serotonin or dopamine [85,86]. Recently, Moon et al. [87] published a review on the use of conducting polymers for the detection of neurotransmitters. This discussed in detail the way in which different systems can be adapted for the different structures of neurotransmitters, such as amino acids, biogenic amines, acetyl choline, and soluble gases. As such, this review will primarily aim to focus on the more recent publications that have not been covered in the mentioned work.

Glassy carbon electrodes (GCEs) are one of the most commonly used working electrodes in electrochemistry. They have been used extensively for the formation of eMIP sensing platforms for small molecules. The most common way to improve an eMIP sensor performance is through modification of the electrode. Zaidi et al. [88] drop-cast reduced graphene oxide (RGO) onto a GCE in the development of an adrenaline sensor using nicotinamide as the functional monomer. The RGO provides a large surface area, extremely low noise, and excellent electrical and thermal conductivity. Nicotinamide is the main constituent of the vitamin B complex and can therefore be considered more environmentally friendly than many other toxic and mutagenic monomers. Using this eMIP based platform with RGO, they managed to produce a limit of detection (LOD) of 3 nM, which compares favourably with other adrenaline sensors found in the literature. An alternative to graphene modified electrodes is using two dimensional hexagonal boron nitride nanosheets (2D-hBN), commonly known as white graphene. This offers important properties, such as chemical and physical stability, increased surface area, good conductivity, and a large band gap, allowing for good transport properties in the visible and IR regions [89]. White graphene was used in conjunction with graphene quantum dots by Yola et al. [90] to form a sensing platform for the detection of serotonin via electropolymerized polyphenol with a low detection limit of 0.2 pM. A common trend in electrode modification used for the deposition of eMIPs is the increase in electrode surface area. This was achieved by Zhang et al. [91] by first covering the GCE surface with chitosan, which allows for the attachment of the negatively charged nanotubes. In this work, the non-conductive eMIP, 3-aminophenylboronic acid, was chosen due to the ability for the boronic acid groups’ ability to interact with the template, which improved the site accessibility and binding affinity for epinephrine compared to earlier work [92,93]. Polymer nanowires can be used to increase the specific surface area and conductivity of the sensor probe. Teng et al. [94] utilised the conducting polymer PPy as nanowires (PPyNWs) to form a foundation for the eMIP (Figure 3). The PPy nanowire was chosen for this application over polyaniline as aniline exhibits a pair of redox peaks in the same potential window as the target, dopamine. This increased conductivity offered through the polymer nanowires allowed for an LOD of 33 nM to be achieved.

Screen-printed electrodes (SPEs) are cost-effective, highly reproducible, can measure small sample volumes, and the technique is inherently suitable for mass-production, thereby, making those electrodes attractive platforms for the functionalization of MIPs with electrochemical methods [95,96]. Many electrode modifications that have been highlighted previously for glassy carbon electrodes can be utilized for the enhancement of screen-printed carbon electrodes (SPCE). Nguy et al. [97], Figure 4, used the formation of a self-assembled monolayer (SAM) on top of the AuNPs to produce a single monolayer of 4-aminophenol. This promoted a more homogeneous growth of the sarcosine imprinted eMIPs and the platform was then utilized as an impedimetric sensor for sarcosine with an LOD of 8.5 nM.

Raj et al. [98] used an SPCE modified with graphene as a basis for a sensing platform for the dual detection of dopamine and 5-hydroxytryptamine (5-HT). 4-amino-3-hydroxyl-1-naphthalene sulfonic acid (AHNSA) was used as the chosen polymer to overcome issues with the aggregation of graphene sheets and offers enhanced stabilization and sensitivity. The π-π stacking and synergistic interaction between the polymer and graphene caused a large net stabilization, shorter ion diffusion paths, and encouragement of electron transport. For both of the target neurotransmitters, the sensing platform produced an enhanced electrocatalytic activity, causing an increase in the peak currents and shifts in potential, managing an LOD of 2 and 3 nM for dopamine and 5-HT, respectively. These findings are summarized in Table 2.

### 2.2. Legal Drugs/Antibiotics

This section will focus on sensors for legal drugs that are developed using electropolymerization. MIPs have been used for the separation, detection, and identification of drugs [99,100,101], with a comprehensive review of the electrochemical methods for the detection of drugs of abuse produced by Florea et al. in 2018 [102]. This section will be divided into the electrode that is employed for polymerization and will describe screen-printed electrodes (SPEs), gold electrodes, and graphene composite materials.

To increase the stability of the SPEs, the sensors are often pretreated by sweeping them between certain potentials in a mild sulfuric acid buffer until reproducible voltammograms are obtained. Pellicer et al. reported on MIPs using Ni (II)-phtalocyanine, a known surface modifier with excellent conducting properties, for the voltammetric detection of the pesticide, fenitrothion [103]. A highly selective sensor for the veterinary drug, oxfendazole, was developed by the direct electrodeposition of pyrrole onto SPEs [104]. The sensitivity of the MIP-modified SPE sensors can be enhanced by including fillers, such as conductive polymers, graphene, and AuNPs [105]. Dechtitrat et al. determined salbutamol levels in food samples and were able to achieve detection limits of 100 pM in buffered solutions using MIP layers into which graphene/PEDOT:PSS layers were inserted, Figure 5.

A selective sensor for the antibiotic, tetracycline, was established with SPEs containing an imprinted polymer layer onto which AuNPs were deposited to enhance the conductivity [106]. SPEs modified with mesoporous carbon material and gold nanoparticles showed a higher electron-transfer rate and electroactive surface compared to bare electrodes, enabling the detection of the feed additive, ractopamine (banned in most countries), in the femtomolar range [107].

Direct deposition of MIPs composed of poly(o-phenylenediamine) (PoPD) on gold electrodes has been employed for the preparation of sensors with high selectivity for furosemide [108] and atrazine [109]. Galvanostatic deposition of pyrrole in the presence of the antibiotic, sulfadimethoxine, was extensively studied by Turco et al. [110]. This research demonstrated that the sensor performance is strongly affected by the electrolyte and overoxidation conditions.

The sensor sensitivity can be enhanced by including support materials on gold electrodes, such as hollow nickel nanospheres [111] or gold nanoparticles (AuNPs) [112]. The latter can be performed by either direct electrodeposition of MIP layers on the surface followed by enrichment of the layers with AuNPs [113] or the simultaneous electropolymerization and AuNPs functionalization. The last method has the advantage of preventing the AuNP from aggregating, which significantly increased the number of binding sites on the surface and led to superior sensor sensitivity [114]. The surface can also first be modified with an SAM, which will promote the subsequent binding of functionalized Au NPs and an electropolymerized imprinted film to form a network-type structure. The sensors prepared according to this strategy had an excellent limit of detection (~3 fM), superior selectivity, and were able to measure the anti-neoplastic drug, gemcitabine, in spiked serum samples [115].

A paper-based analytical device has been developed for the determination of pentachlorophenol based on the response in the photocurrent. To this end, gold nanoparticles were grown onto cellulose fibers into which zinc nanospheres were fixated that were functionalized with a molecularly imprinted polypyrrole film [116].

Graphene, due to its high surface to volume ratio, high conductivity, and high electron mobility, is an interesting material to use for sensor platforms. However, most reports in the literature focus on modified graphene or graphene oxide to facilitate attachment of the polymer. Graphene doped with nitrogen and sulfur moieties was modified with an *o*-aminophenol layer imprinted with the drug, cyclophosphamide. After removal of the template with a mild base, it was used to measure cyclophosphamide levels in rabbits to demonstrate its use for in-real time therapeutic drug monitoring [117]. There are reports in the literature on direct electropolymerization of *o*-phenylenediamine on reduced graphene oxide, which allowed the development of sensors for the selection detection of antibiotics, including chlortetracycline (CTC) [118]. Sun et al. deposited pyrrole on glassy carbon electrodes modified with graphene oxide using CV, which led to sensors that were able to detect quercetin in the nanomolar range [119]. Selective determination of the insecticide, imidacloprid, in real samples was enabled with a poly(*o*-phenylenediamine) polymer imprinted layer functionalized onto a graphene oxide layer [120].

Other strategies to increase the surface area include the use of decorated nanoparticles. Boughrini et al. developed sensors by electropolymerizing *p*-aminothiophenol on gold nanoparticle composites. These sensors were characterized by linear sweep voltammetry and were able to detect the antibiotic, tetracycline, with a very low detection limit (0.22 fM) [121].

The antibiotic, metronidazole, was detected in the femtomolar range using nanoporous gold leaves to enhance the biosensor signal due its high electric conductivity. Proof-of-application was provided by measuring fish tissue samples, which showed the potential of monitoring this antibiotic in biological samples [122]. Another approach to increase the surface area was reported by Jafari et al. [123], who used graphene oxide combined with gold nano-urchins to produce high sensitivity (LOD < 1 nM) sensors for the antibiotic, azithromycin. The use of graphene can be combined with metallic nanoparticles by modifying the hydrophobic graphene surface with a polyelectrolyte, which allows its further decoration with AuNPs. According to this method, MIP-based electrochemical sensors have been developed for the antibiotic, levofloxacin [124].

Carbon nanotubes and (CNTS) and multi-walled carbon nanotubes (MWCNTs) have also been employed to produce MIP-based sensors, since decoration with carbon nanotubes has proven to enhance the electrical signal response. A review on the rise of hybrid MIP-CNTs sensors was described by Dai et al., in 2015, covering various functionalization procedures, including electropolymerization [125]. Here, we will limit ourselves to the use of electropolymerization for the development of drug sensors and provide strategies to enhance the sensor response. Yuan et al. deposited MWCNTs onto glassy carbon electrodes functionalized with a thin polydopamine layer (~15 min) imprinted for the antibiotic, metronidazole. This hybrid sensor was capable of detecting the antibiotic at relevant drug concentrations even in complicated matrices, such as fish tissue [126]. Functionalized carboxylic acid MWCNTs were modified with an imprinted pyrrole layer to develop sensors for the anti-diuretic drug, triamterene, with similar limits of detection as conventional techniques [127]. MWCNTs can also be combined with gold nanoparticles to further enhance the conductivity and sensitivity of the developed assay, which was used for the detection of the anti-viral drug, ganciclovir, in serum samples [128] and the growth promotor, Olaquindox, in feed samples [129]. It was also described that the combination of these two materials leads to faster binding kinetics and superior selectivity over MIP-based sensors that are composed of bulk particles [130]. Electropolymerization of a supramolecular complex was performed on CNTs coupled to gold-coated magnetite, which improved the amperometric determination of pesticide levels in vegetable samples [131]. Furthermore, there is also a report in the literature outlining the use of carbon nitride nanotubes electropolymerized with MIP layers for the detection of pesticides [132,133]. These findings are summarized in Table 3.

### 2.3. Proteins

The detection of proteins introduces new and more complex challenges to the utilization of MIPs in sensing platforms. The complexity, conformational flexibility, solubility, poor mass transfer properties, and permanent entrapment are all issues to be overcome. The design of general MIPs for protein detection is discussed thoroughly in a review by Whitcombe et al. [138], with a more direct focus on the problems encountered with eMIPS in a more recent review by Erdőssy et al. [139] in 2016. As such, this section will focus on the developments not included in that review.

The development of sensing platforms for the detection of proteins focusses on biomarkers for diseases, such as cancer or acute myocardial infarction (AMI). Cardiac troponin is the gold standard of cardiac biomarkers. Elevated levels are present in the blood within 4-6 h following cardiac incident and can remain for up to 10 days [140]. The development of a sensing platform for cardiac troponin utilising eMIPs is a popular route of research. The majority of the work in this area tends towards using SPEs due to the compatibility between the cheap and disposable nature of the SPEs and point-of-care devices. This can be seen in the work from Silva et al., who developed an eMIP based sensing platform based on polypyrrole [141]. This work uses a mixture of monomers to optimize the polymer for the troponin template in the same way as Kim et al. did previously for theophylline [142]. The use of pyrrole-3-carboxylic acid allows for increased binding to the template through the carboxyl groups without interfering with the electropolymerization process that predominantly occurs in the 2 and 5 position of the pyrrole structure. In addition to this, reduced graphene oxide (RGO) was integrated between the SPE and polymer layer to increase the conductivity and improve electron transfer. This combination of graphene and conducting polymer allowed the sensing platform to reach very low detection limits (0.006 ng mL^−1^). MIPs provide excellent selectivity, however, very low detection limits, such as these, are normally reflections of the read-out technology used. Increasing the conductivity and improving electron transfer is important when using a non-conductive polymer, such as *o*-phenylenediamine. Shumyantseva et al. [143,144] achieved this in a myoglobin sensing platform through the inclusion of multi-walled carbon nanotubes (MWCNT), giving a detection limit of 9 ng mL^−1^. MWCNTs can also be mixed with conducting polymers to form composites that allow for the immobilization of sensing components to produce stable and sensitive probes [145]. The increased surface area of the transducer allows for a larger number of imprints and therefore an increased signal in the absence of the template. This followed previous work that deposited polyphenol onto carbon nanotubes for the detection of human ferritin and E7 protein [146]. Polyphenol is another non-conducting polymer that can be used as an effective eMIP due to the π-π stacking interactions it can form with templates. This was used for a myoglobin sensing platform by Ribeiro et al. [147] and a HER2-ECD (breast cancer biomarker) sensor by Pacheco et al. [148] through the utilisation of gold screen printed electrodes (AuSPEs) with a detection limit of 1.6 ng L^−1^. These electrodes can provide a much better surface for adhesion to some polymers, such as 2-aminophenol, which was used as the backbone of an eMIP for the breast cancer biomarker, CA 15-3 [149]. In contrast to this, certain polymers are hard to form on metal surfaces due to poor adhesion between the substrate and the growing phases of the polymer. Ribeiro et al. [150] utilized the formation of SAMs with a glutaraldehyde linker to overcome this problem in an eMIP sensing platform for the same target, CA 15-3 (Figure 6). Poly(toluidine blue), as with other poly(phenazines), is a conductive polymer that offers enhanced electrocatalytic features, can retain redox activity, is water-soluble, and can act as an electrochemical mediator for enzymatic electrodes [151,152,153]. It has proven to be a popular electropolymer for protein platforms due to the plethora of electrostatic interactions and hydrogen bonds that can from between the cationic dye and carbohydrates that comprise proteins.

Another popular electropolymer for the development of eMIPs for proteins is scopoletin, which has been used for transferrin [154], Cytochrome P450 [155], and laccase [156]. Stojanovic et al. used Scopoletin to form an insulating polymer layer imprinted with human serum albumin (HSA) on a gold electrode [157]. Due to the insulating nature of the polymer, detection of the target was achieved through cyclic voltammetry of a potassium ferrocyanide redox marker. Upon re-binding of the HSA target molecules, the imprints become blocked, in turn, blocking the electron transfer to the redox marker.

In addition to electrode surface modification and the formation of standard eMIPs, these processes can be combined. For example, Tamboli et al. [158] first bound an aptamer-protein complex to an Au electrode surface, followed by electropolymerization with polydopamine. This hybrid receptor combined both the established affinity of the aptamer for the template alongside the polymer structure to display superior binding characteristics. These findings are summarized in Table 4.

### 2.4. Bacteria/Viruses

Imprints of larger biological species, such as bacteria and viruses, have also been achieved through the utilisation of eMIPs. These species create unique obstacles to imprinting since they are much larger than proteins, with the majority of viruses between 30 to 700 nm in diameter and bacteria typically between 0.2 to 10 μm, but reaching up to 750 μm [165,166]. Additionally, they each have a complex and unique surface chemistry that can vary from cell to cell, let alone species to species. This requires specific antibodies to be introduced into biosensors to detect these microbes qualitatively and quantitatively.

Electropolymerization techniques possess properties that can directly address each of these issues, by controlling the layer thickness [7] and the direct doping of bacteria into the polymer matrix, leading to the generation of high affinity binding sites [167]. The negatively charged nature of the bacterial cell walls [168] helps to promote the polymerization around them [169]. Furthermore, strong control of the layer thickness is possible, which helps envelope the cells without fully encapsulating them.

One of the first uses of eMIPs for the detection of bacteria utilized the direct deposition of a polypyrrole layer. This was imprinted with the gram-negative bacteria, *P. aeruginosa*, onto a gold evaporated quartz crystal microbalance electrode [170], Figure 7. This system utilized the compatibility between the pyrrole and the anionic surface characteristics of the bacteria. The negative charges present on the cell wall of the bacteria from the presence of phosphate, hydroxyl, and carboxylate groups allow for the expulsion from the eMIP through over-oxidization of the polypyrrole. This was done following treatment with lysozyme, known to digest the cell wall of bacterium [171], and the surfactant, Triton X, to remove the strong interactions between the polysaccharides and polymer. This platform achieved a linear detection range of 10^3^–10^9^ CFU/mL in sterilized water; however, this dropped to 10^7^–10^9^ CFU/mL in apple juice samples.

Instead of developing an eMIP for the entirety of the bacteria, they can be designed to detect specific moieties released by the bacteria. Jiang et al. [172] produced an eMIP for Aflatoxin B1, which is produced by *Aspergillus flavus* and *A. parasiticus*. This used *p*-aminothiophenol as the monomer of choice. In this way, a self-assembled monolayer of the monomer was formed on the Au electrode surface. Following this, the eMIP is formed in the presence of aflatoxin B1 with AuNP modified *p*-aminothiophenol. The re-binding of aflatoxin to the polymer network allows for π-π stacking that results in an increase in conductivity.

The plethora of functional groups typically found on the different monomers used for the creation of MIPs can result in the non-specific adsorption of species other than the target onto the sensor surfaces. Polydopamine has the ability to resist this phenomenon and Chen et al. [173] utilized this on a glassy carbon electrode for the detection of *Escherichia coli* O157:H7. Upon the re-binding of the target in the polymer, the corresponding polyclonal antibody labelled with nitrogen-doped graphene quantum dots (N-GQD) was added. The electrochemiluminescence from the bioconjugation was measured, producing a working range of 10^1^–10^7^ CFU/mL and an LOD of 8 CFU/mL.

As well as specific chemical functional groups, there are also surface proteins on the cell wall of bacteria. Protein A is found as a constituent in 99% of *Staphylococcus aureus* cell walls. Khan et al. [174] used screen-printed carbon electrodes modified with single-walled carbon nanotubes along with an eMIP made from aminophenol. This polymer is a non-conducting polymer and, therefore, produces a distinct layer thickness on an electrode surface. It also contains functional groups favourable for protein attraction, which led to the use of proteinase K for template removal. This platform exhibited changes in the non-imprinted polymer after being exposed to the same template removal strategies as the MIP, thought to be due to peptide bond cleavage from the proteinase K. It also noted an increase in the charge transfer resistance after the template removal, which is unusual. This is proposed to be due to the positive protonation of the protein, increasing the surface concentration of the iron redox probe through ionic interactions.

In addition to surface proteins, some bacteria have flagellar fragments located on their surface, such as *Proteus mirabilis*. In this work, Khan et al. [175] again used the SWCNTs-SPCEs along with a homemade carbon printed electrode. This was formed through the filter paper being coated with hydrophobic paraffin wax and then manually printing the carbon ink onto this substrate. The eMIP was formed on this platform using polyphenol mixed with the template. This platform gave an LOD of 0.6 ng/mL and showed negligible interference from globular proteins or flagellar filaments of other bacteria.

Alternatively, some bacteria, such as *Bacillus anthracis*, are able to survive in harsh conditions and produce spores when they undergo stress. As a simulation for this, Ait Lahcen et al. [176] produced a polypyrrole based eMIP sensing platform for *Bacillus cereus* on a carbon paste electrode. An increased current intensity was found when a layer of polypyrrole was deposited prior to the addition of the target and further electropolymerization. In this work, a concentration of 10^4^ CFU/mL of the target was used for the eMIP formation and hence a linear working range of 10^2^–10^−5^ CFU/mL was achieved.

Although there has been some work on the detection of viruses using MIPs [177,178,179], there has been much less research published on the detection of viruses using eMIPs. This is predominantly due to their flexible 3-D structures, which can fluctuate under minimal interference [180]. Babamiri et al. [181] produced one for the detection of the *HIV-1* gene. This utilized an *o*-phenylenediamine polymer on an Indium Tin Oxide (ITO) electrode surface. This sensing platform used electrochemiluminescence as a detection system by utilizing EuS nanocrystals attached to polyacrylic acid, achieving a working range of 3 fM to 0.3 nM with an LOD of 0.3 fM. Wankar et al. [182] produced an eMIP based platform for the detection of tobacco necrosis virus (TNV) using 200 nm nanofilms of polythiophene. The ~20 nm imprints in the polymer film were verified via Atomic Force Microscopy (AFM). This sensing platform utilized fluorescence spectroscopy at 410 nm with an LOD of 2.29 ng L^−1^. These findings are summarized in Table 5.

## 3. Conclusions

In this review, focus has been applied to the ever-growing field of electrosynthesized molecularly imprinted polymers (eMIPs) and their utilization toward the detection of biologically important moieties. The use of charged monomers can have the benefit of forming strong electrostatic interactions between the monomer and template, but key challenges remain, including undesired high aspecific binding and the ability to control the polymer surface architecture.

Currently, there is a large amount of research published on the detection of small molecules, such as neurotransmitters and drugs, but less on larger macromolecules, such as proteins, bacteria, and viruses. The latter could be due to solubility issues since biomacromolecules tend to dissolve best in aqueous solutions, and water in turn interferes with the interactions between the monomer and template, and the complexity and flexibility of their 3D-structure. Therefore, much of the papers report on chemicals or virulence factors secreted by the biomacromolecules to determine their presence.

However, we expect an increase in the numbers of papers in the field due to novel approaches focusing either on epitope imprinting or on the development of novel monomers that can complexate in aqueous environments. The use of electropolymerization is particularly suitable for these types of sensors since it allows precise control over the morphology, is fast, and seems to have minimal effects on the viability and surface architecture of biomacromolecules.

To improve the response of sensors based on eMIPs for small molecules, the focus tends to be on the modification of electrode materials, such as increasing the surface area by incorporating nanoparticles or graphene sheets. However, some systems have looked at the incorporation of mixed compositions of polymers and new monomers to produce more selective, sensitive, and ‘green’ systems. In the upcoming years, more reports are expected in the field of both novel functional and crosslinker monomers, since the latter can also have a significant impact on the binding properties of the developed sensor platforms.

Molecularly imprinted polymers have been shown to be adept as recognition elements in biosensing systems and there is a clear trend to move away from natural recognition elements due to their high-cost and limited stability. The incorporation of electropolymerization into the synthesis of eMIPs allows for direct and strong adherence to the transducer of the sensor, an increased speed of preparation, and control of the layer thickness and morphology. Therefore, it is expected that the area of eMIP research will continue to grow in the coming years.

## Figures and Tables

**Figure 1 sensors-19-01204-f001:**
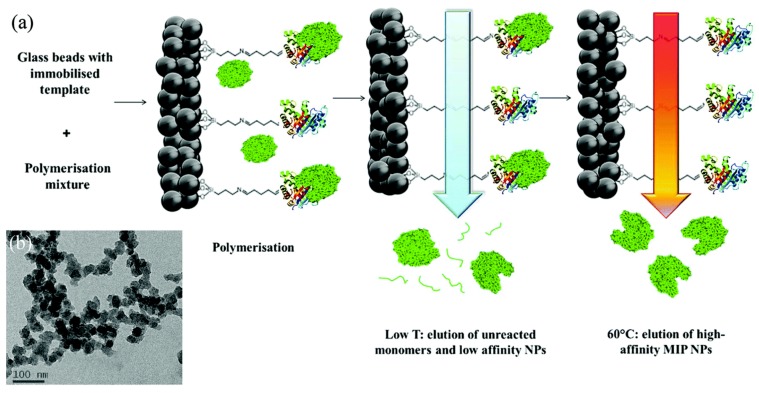
(**a**) Scheme of the solid-phase synthesis of nanoMIPs. In this example, a protein is shown as the template molecule. (**b**) Representative TEM of biotin MIPs. Reproduced from [31]—Published by The Royal Society of Chemistry.

**Figure 2 sensors-19-01204-f002:**
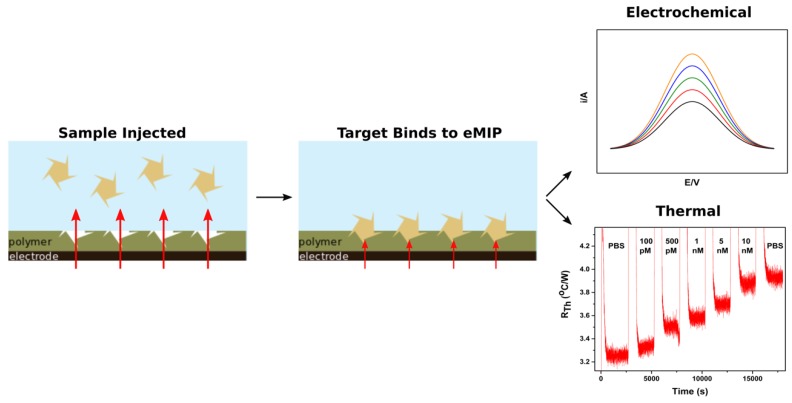
Schematic of the typical development of an eMIP based sensing platform with two examples of detection read out. The template is first mixed with the monomer solution, then using either the constant potential or cyclic voltammetry. The template is then removed from the polymer layer. Upon the addition of sample, the target rebinds to the polymer layer, which can be measured through various techniques, such as voltammetry or thermal analysis.

**Figure 3 sensors-19-01204-f003:**
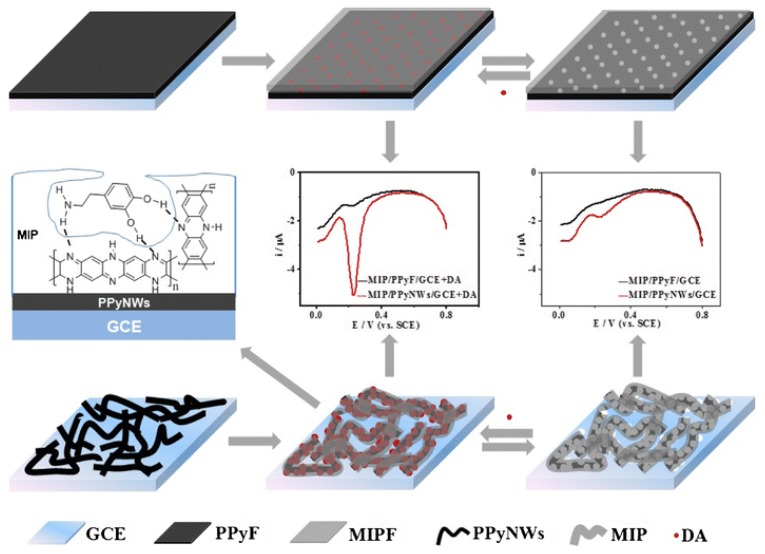
Schematic illustrations of the fabrication procedure of MIP/PPyNWs/GCE (Differential pulse voltammograms: Slight current of dopamine (DA)can be seen in left figure due to the extraction of DA; MIP/PPyNWs/GCE showed much higher current of DA than that of MIP/PPyF/GCE in middle figure owing to the excellent electrocatalysis of PPyNWs to DA. Reproduced by permission from Springer Nature, Microchimica Acta, Teng et al. [94], Copyright 2017.

**Figure 4 sensors-19-01204-f004:**
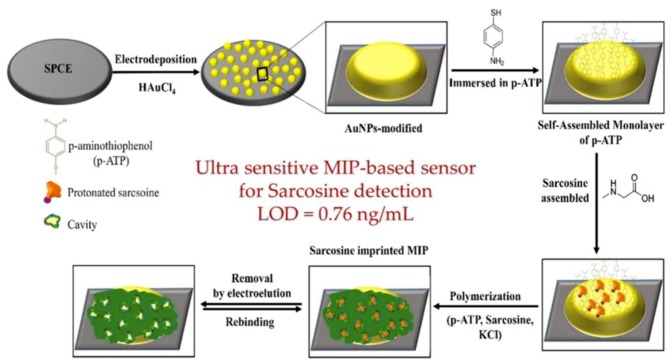
Schematic diagram of the sensing platform produced by Nguy et al. utilising a screen-printed carbon electrode (SPCE) functionalized with AuNPs and poly(4-aminophenol) imprinted with sarcosine. Reprinted from Sensors and Actuators B: Chemical, 246, Nguy et al., Development of an impedimetric sensor for the label-free detection of the amino acid sarcosine with molecularly imprinted polymer receptors., 461–470, 2017, with permission from Elsevier.

**Figure 5 sensors-19-01204-f005:**
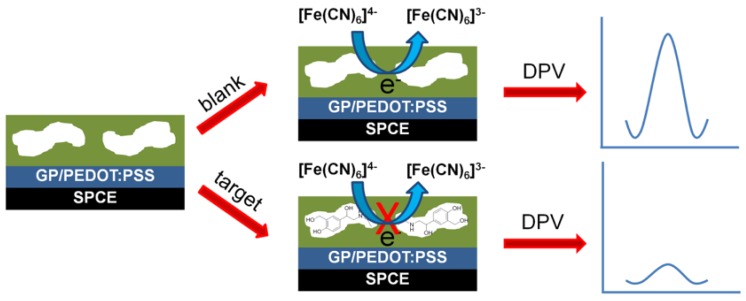
Schematic illustration of the electrochemical determination of salbutamol using an eMIP sensing platform. Reproduced from [105]—Published by The Royal Society of Chemistry.

**Figure 6 sensors-19-01204-f006:**
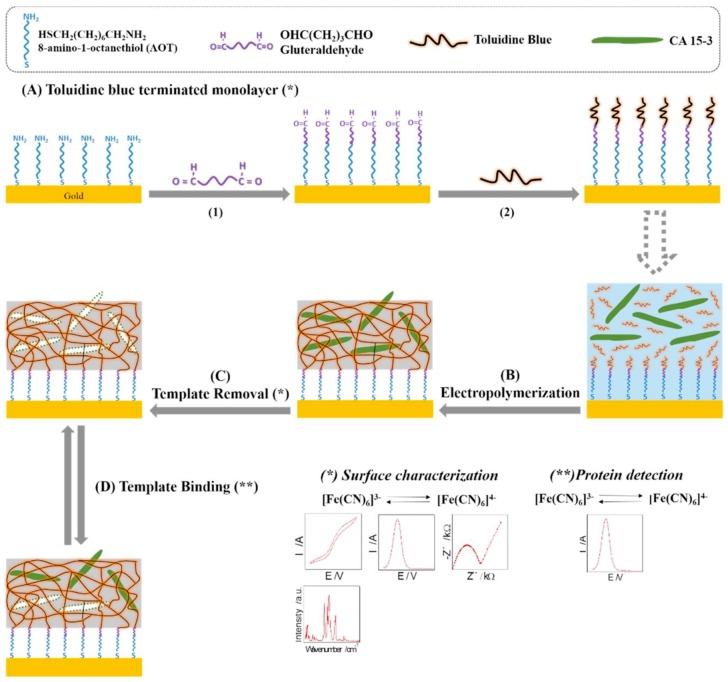
Schematic representation of the sensing platform produced by Ribeiro et al. for the detection of the cancer biomarker, CA 15-3. A Self-Assembled Monolayer (SAM) is formed on the Au electrode with a glutaraldehyde linking this to a layer of the monomer. Reprinted from Biosensors and Bioelectronics, 109, Ribeiro et al., Disposable electrochemical detection of the breast cancer tumour marker, CA 15-3, using poly(Toluidine Blue) as the imprinted polymer receptor, 246-254, 2018, with permission from Elsevier.

**Figure 7 sensors-19-01204-f007:**
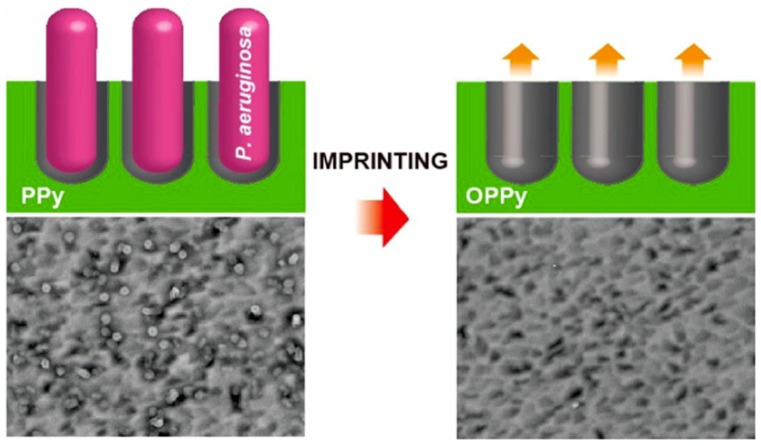
Figure showing (left) the formation of the polypyrrole eMIP with *P. aeruginosa* as a template and (right) the eMIP layer after the expulsion of the bacteria through treatment with lysozyme, Triton X, and over-oxidization. Reprinted with permission from [170]. Copyright 2013 American Chemical Society.

**Table 1 sensors-19-01204-t001:** Commonly used monomers in the formation of eMIPs alongside their structures and selected references utilising them as eMIPs.

Monomer	Monomer Structure	Reference
Pyrrole	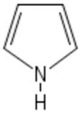	[64,65,66]
o-phenylenediamine	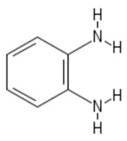	[67,68,69]
Aniline	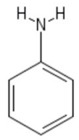	[70,71,72]
Phenol	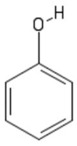	[73,74,75]
Carbazole	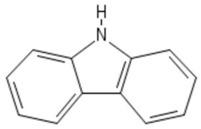	[76,77,78]
Scopoletin	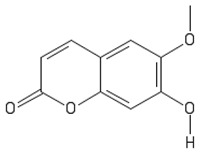	[79,80,81]

**Table 2 sensors-19-01204-t002:** Summary of recent publications for the detection of neurotransmitters using eMIPs. Includes information on the functional monomer used, electrode material, polymerization technique, detection method, and Limit Of Detection (LOD).

Template	Functional Monomer	Electrode Material	EP Technique	Detection Method	LOD	Ref.
epinephrine	nicotinamide	GCE^a^/rGO^c^	CV^k^	DPV^m^	3 nM	[88]
serotonin	phenol	GCE^a^/GQDs^d^/2D-hBN^e^	CV^k^	DPV^m^	0.2 pM	[90]
epinephrine	3-aminophenylboronic acid	GCE^a^/MWCNTs^f^	CV^k^	DPV^m^	35 nM	[91]
epinephrine	2,4,6-trisacrylamido-1,3,5-triazine	Au	CV^k^	DPV^m^	12 nM	[93]
dopamine	*o*-phenylenediamine	GCE/^a^PPyNWs^g^	CV^k^	DPV^m^	33 nM	[94]
dopamine/serotonin	4-amino-3-hydroxy-1-napthalenesulfonic acid	SPCE^h^/GR^i^	CV^k^	SWV^n^	2 & 3 nM	[98]
Sarcosine	*p*-aminothiophenol	SPCE^h^/AuNPs^j^	CV^k^	EIS^l^	1 nM	[97]

^a^ GCE—Glassy Carbon Electrode; ^b^ GO—Graphene Oxide; ^c^ rGO—reduced Graphene Oxide; ^d^ GQDs—Graphene Quantem Dots; ^e^ 2D-hBN—Two Dimensional hexagonal Boron Nitride; ^f^ MWCNTs—Multi-walled Carbon Nanotubes; ^g^ PPyNWs—Polypyrrole Nanowires; ^h^ SPCE—Screen-Printed Carbon Electrode; ^i^ GR—Graphene; ^j^ AuNPs—Gold Nanoparticles; ^k^ CV—Cyclic Voltammetry; ^l^ EIS—Electrochemical Impedance Spectroscopy; ^m^ DPV—Differential Pulse Voltammetry; ^n^ SWV—Square Wave Voltammetry.

**Table 3 sensors-19-01204-t003:** Summary of recent publications for the detection of drugs using eMIPs. Includes information on the functional monomer used, electrode material, polymerization technique, detection method, and LOD.

Template	Functional Monomer	Electrode Material	EP Technique	Detection Method	LOD	Ref.
oxfendazole	pyrrole	SPCE^a^	CV^m^	DPV^o^ /SWV^p^	10/8 μg/kg	[104]
salbutamol	3-aminophenylboronic acid/*o*-phenylenediamine	SPCE^a^/GR^b^/PEDOT:PSS^c^	CV^m^	DPV^o^	0.1 nM	[105]
naloxone	4-aminobenzoic acid	SPCE^a^/MWCNT^d^	CV^m^	DPV^o^	0.2 μM	[134]
tetracycline	pyrrole	SPCE^a^/AuNPs^e^	CV^m^	DPV^o^	0.65 μM	[106]
ractopamine	*p*-aminothiophenol	SPCE^a^/OMC^f^/AuNPs^e^	CV^m^	DPV^o^	42.3 pM	[107]
furosemide	*o*-phenylenediamine	Au	CV^m^	DPV^o^	70 nM	[108]
atrazine	*o*-phenylenediamine	Au	CV^m^	DPV^o^	1 nM	[109]
sulfadimethoxine	pyrrole	Au	GD^n^	CA^q^	70 μM	[110]
tolazoline	*o*-aminothiophenol	Au	CV^m^	CV^m^	0.016 μg/mL	[113]
gemcitabine	*p*-aminothiophenol	Au/AuNPs^e^	CV^m^	LSV^r^	3 fM	[115]
tetracycline	*p*-aminothiophenol	Au/AuNPs^e^	CV^m^	LSV^r^	0.22 fM	[121]
metronidazole	*o*-phenylenediamine	Au/NPGL^g^	CV^m^	CV^m^	18 pM	[122]
azithromycin	aniline	GCE^h^/GO^i^/GNU^j^	CV^m^	DPV^o^	0.1 nM	[123]
levofloxacin	pyrrole	GCE^h^/G-AuNPs^k^	CV^m^	DPV^o^	0.53 μM	[124]
metronidazole	dopamine	GCE^h^/MWCNT^d^	CV^m^	CV^m^	49.2 ng/L	[126]
triamterene	pyrrole	PGE^l^/MWCNT^d^	CV^m^	DPV^o^	3.35 nM	[127]
ganciclovir	2,2′-dithiodianiline	GCE^h^/MWCNT^d^/AuNPs^e^	CV^m^	DPAS^s^	1.5 nM	[128]
methimazole	pyrrole	PGE^l^	CV^m^	DPV^o^	3 μM	[135]
β-estradiol	3,6-diamino-9-ethylcarbazole	GCE^h^	CV^m^	EIS^t^	0.36 aM	[136]
testosterone	*o*-phenylenediamine	GCE^h^/GO^i^	CV^m^	EIS^t^	0.4 fM	[137]

^a^ SPCE—Screen-Printed Carbon Electrode; ^b^ GR—Graphene; ^c^ PEDOT:PSS—poly(3,4-ethylenedioxythiophene) polystyrene sulfonate; ^d^ MWCNT—Multiwalled Carbon Nanotubes; ^e^ AuNPs—Gold Nanoparticles; ^f^ OMC—Ordered Mesoporous Carbon; ^g^ NPGL—Nanoporous Gold Leaf; ^h^ GCE—Glassy Carbon Electrode; ^i^ GO—Graphene Oxide; ^j^ GNU—Gold Nano Urchins; ^k^ G-AuNPs—Graphene-Gold Nanoparticles; ^l^ PGE—Pencil Graphite Electrode; ^m^ CV—Cyclic Voltammetry; ^n^ GD—Galvanostatic Deposition; ^o^ DPV—Differential Pulse Voltammetry; ^p^ SWV—Square Wave Voltammetry; ^q^ CA—Chronoamperometry; ^r^ LSV—Linear Sweep Voltammetry; ^s^ DPAS—Differential Pulse Absorption Stripping; ^t^ EIS—Electrochemical Impedance Spectroscopy.

**Table 4 sensors-19-01204-t004:** Summary of recent publications for the detection of proteins using eMIPs. Includes information on the functional monomer used, electrode material, polymerization technique, detection method, and LOD.

Table	Functional Monomer	Electrode Material	EP Technique	Detection Method	LOD	Ref.
troponin T	pyrrole/pyrrole-3-carboxylic acid	SPCE^f^	CV^i^	DPV^k^	0.006 ng/mL	[141]
myoglobin	*o*-phenylenediamine	SPCE^f^	CV^i^	DPV^k^	0.5 nM	[143]
hemoglobin	TBA^e^	Au	CV^i^	CA^l^	82 nM	[145]
HER2-ECD^a^	phenol	AuSPE^g^	CV^i^	DPV^k^	1.6 ng/mL	[148]
CA 15-3^b^	*o*-aminophenol	AuSPE^g^	CV^i^	DPV^k^	1.5 U/mL	[149]
CA 15-3 ^b^	toluidine blue	AuSPE^g^	CV^i^	DPV^k^	0.1 U/mL	[150]
PSA^c^	dopamine	Au	CV^i^	MOSFET^m^	0.1 pg/mL	[158]
Annexin A3	Caffeic acid	SPCE^f^	CP^j^	SWV^n^	0.095 ng/mL	[159]
S-ovalbumin	pyrrole	GCE^h^	CV^i^	DPV^k^	2.95 × 10^−9^ mg/mL	[160]
Microseminoprotein-beta	Caffeic acid	SPCE^f^	CP^j^	SWV^n^	0.12 ng/mL	[161]
Human serum albumin	Bis(2,2′-bithien-5-yl)methane	Au	CV^i^	DPV^k^	0.25 pM	[162]
CA 15-3^b^	pyrrole	FTO^p^	CV^i^	Potentiometry	1.07 U/mL	[163]
CA 15-3^b^	*o*-phenylenediamine	AuSPE^g^	CV^i^	SWV^n^	0.05 U/mL	[164]
Human serum albumin	scopoletin	Au	CV^i^	CV^i^	56 nM	[157]
transferrin	scopoletin	Au	CV^i^	SWV^n^	-	[154]

^a^ HER2-ECD—Human Epidermal Growth Factor Receptor 2-Extracellular Domain; ^b^ CA 15-3—Cancer Antigen 15-3; ^c^ PSA—Prostate -Specific Antigen; ^d^ CEA—Carcinoembryonic Antigen; ^e^ TBA—2,2′,5,5′-terthiophene-3′-*p*-benzoic acid; ^f^ SPCE—Screen-Printed Carbon Electrode; ^g^ AuSPE—Gold Screen-Printed Electrode; ^h^ GCE—Glassy Carbon Electrode; ^i^ CV—Cyclic Voltammetry; ^j^ CP—Constant Potential; ^k^ DPV—Differential Pulse Voltammetry; ^l^ CA—Chronoamperometry; ^m^ MOSFET—Metal Oxide Semiconductor Field-Effect Transistor; ^n^ SWV—Square Wave Voltammetry; ^o^ EIS—Electrochemical Impedance Spectroscopy; ^p^ FTO—Fluorine doped Tin Oxide.

**Table 5 sensors-19-01204-t005:** Summary of recent publications for the detection of bacteria and viruses using eMIPs. Includes information on the functional monomer used, electrode material, polymerization technique, detection method, and LOD.

Target	Functional Monomer	Electrode Material	EP Technique	Detection Method	LOD	Ref.
*P. aeruginosa*	pyrrole	Au	CP^f^	QCM^h^	10^3^ CFU/mL	[170]
*E. coli*	dopamine	GCE^a^	CV^g^	ECL^j^	8 CFU/mL	[173]
protein A (*S. aureus*)	*m*-aminophenol	SPCE^b^	CV^g^	EIS^k^	0.6 nM	[174]
*P. mirabilis*	phenol	SPCE^b^/HP^c^ CPE^d^	CV^g^	EIS^k^	0.7 mg/mL	[175]
*B. cereus*	pyrrole	CPE^d^	CV^g^	CV^g^	10^2^ CFU/mL	[176]
HIV	*o*-phenylenediamine	ITO^e^	CV^g^	ECL^j^	0.3 fM	[181]
TNV^l^	thiophene	Au	CV	Fluorescence	2.29 ng/L	[182]

^a^ GCE—Glassy Carbon Electrode; ^b^ SPCE—Screen-Printed Carbon Electrode; ^c^ HP CPEs—Hand-Printed; ^d^ CPE—Carbon Paste Electrode; ^e^ ITO—Indium Tin Oxide; ^f^ CP—Constant Potential; ^g^ CV—Cyclic Voltammetry; ^h^ QCM—Quartz Crystal Microbalance; ^i^ LSV—Linear Sweep Voltammetry; ^j^ ECL—Electrochemiluminescence; ^k^ EIS—Electrochemical Impedance Spectroscopy; ^l^ TNV—Tobacco Necrosis Virus.

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
