# Peer review of "Recent Advances in Electrosynthesized Molecularly Imprinted Polymer Sensing Platforms for Bioanalyte Detection"

_sensors, 2019, doi:10.3390/s19051204_

Reviewer 1 Report

The review submitted by R.D. Crapnell et al. describes an overview about the preparation methods of molecularly imprinted polymers and their application in detection of some important classes of compounds like: neurotransmitters, drugs, proteins, viruses and bacteria. It is a good article containing a lot of information, especially in the form of bibliographical references.

The work is well-ordered and focused on a topic of significant relevance from the scientific and practical point of view and general considerations are also pointed out. Therefore I consider this review publishable in ‘Sensors’ after addressing the following concerns:

In table 2, estradiol and testosterone are not neurotransmitters; they should be integrated in table 3 due to the fact that besides being hormones, they are used as drugs in hormonal substitution therapy.

In subchapter 2.2., the review published by Florea et al. (Current Opinion in Electrochemistry, 11, October 2018, 34-40) which is a comprehensive review on electrochemical methods for the detection of drug of abuse should be placed after other examples of MIPs for drugs (medication) and the title should be modified accordingly.

In the same subchapter, fig 5 presents in fact an immunosensor for clenbuterol using MWCNT deposited on SPE-PEDOT and the antibody for clenbuterol is linked after activation with EDC/NHS. It is not an example of MIP based sensors (ref 98). The same observation goes for reference 146 which reports an immunosensor for CEA using pyrrole-3-carboxylic acid as a substrate for antibody immobilization.

I also recommend illustrating the review with some other representative figures for the described methodologies.

Author Response

Reviewer 1:

In table 2, estradiol and testosterone are not neurotransmitters; they should be integrated in table 3 due to the fact that besides being hormones, they are used as drugs in hormonal substitution therapy.

The authors agree with this and therefore the references in question have been removed from table 2 and incorporated into table 3. The text referring to these references have also been removed from the section.

2. In subchapter 2.2., the review published by Florea et al. (Current Opinion in Electrochemistry, 11, October 2018, 34-40) which is a comprehensive review on electrochemical methods for the detection of drug of abuse should be placed after other examples of MIPs for drugs (medication) and the title should be modified accordingly. 

This has been remedied by adding in the following at lines 288-291:

“MIPs have been used for the separation, detection and identification of drugs [100–102]; with a comprehensive review of electrochemical methods for the detection of drugs of abuse produced by Florea et al. in 2018 [103].

3. In the same subchapter, fig 5 presents in fact an immunosensor for clenbuterol using MWCNT deposited on SPE-PEDOT and the antibody for clenbuterol is linked after activation with EDC/NHS. It is not an example of MIP based sensors (ref 98). The same observation goes for reference 146 which reports an immunosensor for CEA using pyrrole-3-carboxylic acid as a substrate for antibody immobilization.

The authors agree with this and have therefore removed Figure 5 along with the reference from the text and table 3. A new Figure 5 has been implemented from reference [106]

The authors agree that reference 146 was not suitable and it has therefore been removed from table 4.

4. I also recommend illustrating the review with some other representative figures for the described methodologies.

Figure 3 has been added which shows the use of PPyNWs.

Reviewer 2 Report

The manuscript might be published after

-elimination of several  fatal errors:

="first development"  of  MIPs was not by Wulff but by Polyakov.

=Tab.2: ß-estradiol and testestorone are no neurotransmitters,

=Tab.5:aflatoxin and proteinA are no bacteria

--a realistic estimate of the (commercial) application of MIPs should be added

=MIPs can not (jet) compete with antibodies in routine applications!!!

="The main commercial application lies in the removal..." Pease give other  commercial applications if there are any!

- many papers cited in this review present non-realistic sensitivities :

 = LLOD in the sub-pM concentration range of MIP-sensors: using a "redox marker"can not be explained by the affinities of MIPs. A  critical comment is required!

-selection of essential papers

-The selection of the cited papers is not representative::

=For example the citations of the Silva group dominate the protein part (without any discussions of the claimed parameters).Why are two similar papers about the same Mb-MIP  in the reference list? Ref.136 is from 2010 and presents non-realistic sensitivities!

=Scopoletin has been ignored in Tab.1 and in the presentation of examples in spite of its successful applications for both low-molecular analytes  and for proteins (transferrin, HSA, Cyt P450,laccase).

Author Response

Reviewer 2:

 ="first development"  of  MIPs was not by Wulff but by Polyakov.

The authors agree with this and have altered the text between line 58 – 60 with:

“Molecular imprinting technology was first developed by Polyakov [10] in the 1930s, with further progression by Wulff [11] in the 1970s but increased in popularity when Mosbach and co-workers [12] introduced the non-covalent imprinting approach in the 1990s.

=Tab.2: ß-estradiol and testestorone are no neurotransmitters,

The authors agree and therefore these have been removed from table 2 and incorporated into table 3 as they are used as drugs in hormonal substitution therapy. The text referring to thee references have also been removed from section 2.1.

=Tab.5:aflatoxin and proteinA are no bacteria

In the work on aflatoxinB1 the authors mention the relationship to Aspergillus flavus and A. parasiticus, but you are right they do not mention using this to detect these bacteria and have been removed from table 5.

The work from Khan et al.  on Protein A detection specifically mentions the future aims to be for the detection of S. aureus due to the presence of Protein A as a cell wall constituent. (S. Aureus) hass been added in brackets to make this more clear.

-a realistic estimate of the (commercial) application of MIPs should be added

The authors agree that this helps add context to the review and have therefore added in the following between lines 144-164:

“There are currently over 1000 patents on molecular imprinting, with a range of commercially available products available for filtration and purification. They have been utilized in chromatography columns due to the simple packing process of the particles. Whereas this is a scalable approach, there is no process in place that can mass-produce MIPs with high affinity into electrodes. Peeters et al. [24] have described a method for the mass-production of MIPs onto Screen-Printed Carbon Electrodes (SPCEs) by direct mixing of particles with the ink. However, these microparticles have limited affinity and only ~30 % of the particles can be taken up in the ink. This may require adaptation of the ink to reflect the nature of the MIP particles; for hydrophilic particles the ink will need to be altered [47]. The use of nanoparticles as developed by technology in Leicester [32,34] opens up new functionalization procedures since they are not cross-linked and therefore water-soluble. They also produces superior affinity since they are pre-selected, through temperature elution, and can be produced in larger volumes. At the present time MIPs are an emerging technology and cannot compete with traditional antibodies. In the future it is expected that this approach is more suitable for sensor applications and have the potential to replace antibodies in standard assays [33,48,49]. Molecular imprinting is an emerging technology and in order to produce polymers with similar affinity as antibodies, more optimization is required per target. Some examples of commercial outputs utilizing MIPs include SupelMIP (Solid Phase Extraction (SPE) cartridges for drugs and pesticides), AFFINIMIPÒ SPE (SPE for contaminants and endocrine disruptors), Biotage (industrial resins to remove trace pesticides), Ligar (recovering target molecules from plants, isolating flavor molecules and extracting contaminants from liquids) and MIPDiagnostics (bespoke nanoparticle synthesis which can be used in various applications including sensing platforms).

=MIPs can not (jet) compete with antibodies in routine applications!!!

The authors agree that at the present time MIPs cannot compete with traditional antibodies. Molecular Imprinting is an emerging technology and in order to produce polymers with similar affinity to antibodies more optimization is required per target. This has been addressed in the text between lines 155-157:

“At the present time MIPs are an emerging technology and cannot compete with traditional antibodies. In the future it is expected that this approach is more suitable for sensor applications and have the potential to replace antibodies in standard assays [33,48,49].

="The main commercial application lies in the removal..." Pease give other  commercial applications if there are any!

The vast majority of commercial applications lie in extraction although there are now examples of bespoke particles that can be commercially bought for use in sensing applications. Some commercial suppliers are outlined between lines 159-164:

“Some examples of commercial outputs utilizing MIPs include SupelMIP (Solid Phase Extraction (SPE) cartridges for drugs and pesticides), AFFINIMIPÒ SPE (SPE for contaminants and endocrine disruptors), Biotage (industrial resins to remove trace pesticides), Ligar (recovering target molecules from plants, isolating flavor molecules and extracting contaminants from liquids) and MIPDiagnostics (bespoke nanoparticle synthesis which can be used in various applications including sensing platforms).

= LLOD in the sub-pM concentration range of MIP-sensors: using a "redox marker"can not be explained by the affinities of MIPs. A  critical comment is required!

We respectfully disagree with the reviewer here. The limit of detection is not just determined by the recognition element (MIPs), but also the read-out technology that is used has a significant influence on the limit of detection. We have added in the following between lines 402-406 to make this more clear:

“This combination of graphene and conducting polymer allowed the sensing platform to reach very low detection limits (0.006 ng mL-1). MIPs provide excellent selectivity however, very low detection limits such as these are normally reflections of the read-out technology used. Increasing the conductivity and improving electron transfer is important when using a non-conductive polymer such as o-phenylenediamine.

=For example the citations of the Silva group dominate the protein part (without any discussions of the claimed parameters).Why are two similar papers about the same Mb-MIP  in the reference list? Ref.136 is from 2010 and presents non-realistic sensitivities!

The authors have added in another part to section 2.3 to broaden the spectrum from lines 409-413:

“Another popular electropolymer for the development of eMIPs for proteins is scopoletin; which has been used for transferrin [154], Cytochrome P450 [155] and laccase [156]. Stojanovic et al. used Scopoletin to form an insulating polymer layer imprinted with Human Serum Albumin (HSA) on a gold electrode [157]. Due to the insulating nature of the polymer, detection of the target was achieved through cyclic voltammetry of a potassium ferrocyanide redox marker. Upon re-binding of the HSA target molecules, the imprints become blocked; in turn blocking the electron transfer to the redox marker.

The two papers that are similar on the reference list for Mb-MIP are due to it being the same group publishing a paper after each stage of their work. They have a paper on the preparation of the system and testing in lab settings and then one applying it to real samples.

The paper from 2010 is only briefly mentioned and is not a focus of the review.

=Scopoletin has been ignored in Tab.1 and in the presentation of examples in spite of its successful applications for both low-molecular analytes  and for proteins (transferrin, HSA, Cyt P450,laccase).

The authors agree that there has been significant work published using scopoletin as a functional monomer for eMIP synthesis. Therefore this has been added to table 1 with references. In addition to this the work on the detection of HSA and transferrin being included in table 4. A section has been added into section 2.3, see above comment.

Reviewer 3 Report

Lines 46-47: What did authors mean by “…inability to simultaneously optimize the conditions for both imprinting and re-binding.”?

Lines 60-61: Bleeding is a common problem for MIPs due to their highly crosslinked nature. Why did it mention to be important just for non-covalent MIPs?

Line 67: For ions as temples, synthesized polymers are termed ion imprinted polymers (IIPs).

Line: 96: What is “impact factor”?

Line 98: Please add crosslinkers into the: “standard commercially available monomers”.

Line 100: Please add the references after “(Figure 1)”.

Lines 111-112: In “The approach by Haupt focuses on the attachment of an affinity ligand of the protein to the solid support, which enables direct immobilization of the protein [34].”, the mentioned reference does not contain what is described. Please check it. References 34 and 35 use relatively different methodologies.

Line 118: what are “ATRP” and “RAFT”? Please describe all abbreviations at the first stage of their use.

Line 124-125: Please add suitable reference for “This approach is not scalable, but leads to high affinity and can be used to integrate sensors into biomedical devices such as titanium implants.”.

Line 146: For the electropolymerization of MIP layers, template molecules must be also available in the solution.

Lines 148: Please replace “an array” with different.

Line149: Please add suitable references for each mentioned technique.

Line162: Figure 2 shows the sensing mechanism of synthesized MIP layer. Please correct the “The formation of eMIPs through electropolymerization occurs predominantly through the same steps as seen in Figure 2.”

Figure 3 can be deleted.

The valuable information which is monomer and the references can be mentioned in a sentence. Please remove table 1.

206-207: The review published by Moon [78] is quite new. How many more new studies have been published which were not mentioned in that review?

Line 210-212: the sentence “There are…. of the MIP.” is confusing. Please correct or rewrite it.

Line 212: The sentence is started with the work from Liu et al. [79]. Immediately,  a new reference is mentiond to give information about carbazole based monomer… [80]. The next sentence is started with “This polymer…”. This polymer refers to which reference? Please consider that through the manuscript to write fluently and with a clear structure and not like a separated puzzle.

Lines 435-436: For “Electropolymerization techniques possess properties that can directly address each of these issues, allowing high affinity binding sites to be generated.” What are the reasons to prove that? References?

Line 436: What is the charged nature of the cell`s surfaces? Please add more information with references.

Author Response

Reviewer 3:

Lines 46-47: What did authors mean by “…inability to simultaneously optimize the conditions for both imprinting and re-binding.”?

The authors meant that it is difficult to optimize the imprinting conditions and re-binding conditions simultaneously due to various factors such as solubility and interactions between the monomer, target and solvents. As such, we have altered the section at lines 46-49 to read:

“However, there are areas in which advancements in electropolymerization need to be made. These include the removal of imprinted templates and inability to simultaneously optimize the conditions for both imprinting and re-binding; such as solubility and minimizing interactions between solvent, target and monomer.

Lines 60-61: Bleeding is a common problem for MIPs due to their highly crosslinked nature. Why did it mention to be important just for non-covalent MIPs?

The authors agree that this poses a significant problem for covalently bound MIPs and have therefore altered lines 62-63 to:

“The latter can be challenging since template leaching is considered a common problem for the use of MIPs, both covalent and non-covalent [13].”

Line 67: For ions as temples, synthesized polymers are termed ion imprinted polymers (IIPs).

The authors agree and as such have added in at line 69-70:

“(known as Ion Imprinted Polymers)

Line: 96: What is “impact factor”?

This was supposed to read as “Imprint Factor” and has such been changed at line 98.

Line 98: Please add crosslinkers into the: “standard commercially available monomers”.

At line 100-101 the following has been added:

“usually standard commercially available monomers and cross-linkers are used.

Line 100: Please add the references after “(Figure 1)”.

Added in the reference corresponding to figure 1 at line 103:

“(Figure 1 [31]).

Lines 111-112: In “The approach by Haupt focuses on the attachment of an affinity ligand of the protein to the solid support, which enables direct immobilization of the protein [34].”, the mentioned reference does not contain what is described. Please check it. References 34 and 35 use relatively different methodologies.

The authors agree and have change the references to what they should have been originally:

For the protein immobilization at line 115:

35. Ambrosini, S.; Beyazit, S.; Haupt, K.; Tse Sum Bui, B. Solid-phase synthesis of molecularly imprinted nanoparticles for protein recognition. Chem. Commun. 2013, 49, 6746, doi:10.1039/c3cc41701h.

For the thermoresponse and in-vivo application at line 117:

31. Canfarotta, F.; Czulak, J.; Betlem, K.; Sachdeva, A.; Eersels, K.; Van Grinsven, B.; Cleij, T.J.; Peeters, M. A novel thermal detection method based on molecularly imprinted nanoparticles as recognition elements. Nanoscale 2018, 10, 2081–2089, doi:10.1039/c7nr07785h.

Line 118: what are “ATRP” and “RAFT”? Please describe all abbreviations at the first stage of their use.

The authors agree and have therefore defined these in the text at lines: 121-123:

“Atom Transfer Radical Polymerization (ATRP) [36], Reversible Addition Fragmentation Chain-Transfer Polymerization RAFT [37],

Line 124-125: Please add suitable reference for “This approach is not scalable, but leads to high affinity and can be used to integrate sensors into biomedical devices such as titanium implants.”.

The authors agree the reference was needed. As such the sentence has been altered slightly at lines 129-130:

“This facile approach is not scalable, but leads to high affinity and can be used to integrate sensors into biomedical devices [39].

With the following reference added:

39. Pan, G.; Zhang, Y.; Guo, X.; Li, C.; Zhang, H. An efficient approach to obtaining water-compatible and stimuli-responsive molecularly imprinted polymers by the facile surface-grafting of functional polymer brushes via RAFT polymerization. Biosens. Bioelectron. 2010, 26, 976–982, doi:10.1016/J.BIOS.2010.08.040.

Line 146: For the electropolymerization of MIP layers, template molecules must be also available in the solution.

The authors agree and have therefore added in at line 168-169:

“Electropolymerization is a deposition procedure in which a conductive polymer layer is formed or coated upon an electrode/supporting substrate material in the presence of the desired template”

Lines 148: Please replace “an array” with different.

This has been replaced at line 174 with:

Electropolymerization can be achieved through a variety of electrochemical techniques

Line149: Please add suitable references for each mentioned technique.

The following references, showing an example of electropolymerization using each technique mentioned, have been added at line 175:

Voltammetric:

51.        Losito, I.; Palmisano, F.; Zambonin, P.G. o-phenylenediamine electropolymerization by cyclic voltammetry combined with electrospray ionization-ion trap mass spectrometry. Anal. Chem. 2003, 75, 4988–4995, doi:10.1021/ac0342424.

Potentiostatic:

52.        Komaba, S.; Seyama, M.; Momma, T.; Osaka, T. Potentiometric biosensor for urea based on electropolymerized electroinactive polypyrrole. Electrochim. Acta 1997, 42, 383–388, doi:10.1016/S0013-4686(96)00226-5.

Galvanostatic:

53.       Uang, Y.M.; Chou, T.C. Criteria for designing a polypyrrole glucose biosensor by galvanostatic electropolymerization. Electroanalysis 2002, 14, 1564–1570.

Line162: Figure 2 shows the sensing mechanism of synthesized MIP layer. Please correct the “The formation of eMIPs through electropolymerization occurs predominantly through the same steps as seen in Figure 2.”

The authors agree and have since change the text at line 186 to:

“The sensing mechanism of eMIPs commonly occurs through the same steps as seen in Figure 2.

Figure 3 can be deleted. 

Figure 3 has been removed.

The valuable information which is monomer and the references can be mentioned in a sentence. Please remove table 1.

The authors do not feel this is necessary. While they agree that this information could be displayed in the text. They feel that showing the structure of the monomers helps the reader to understand and consider the different interactions that could occur between the polymer and template during the electropolymerization process. They therefore feel there is no need to remove table 1.

206-207: The review published by Moon [78] is quite new. How many more new studies have been published which were not mentioned in that review?

The authors agree that this review is recently published. The following papers for references 89, 91, 92, 93, 94, 95 and 98 are not included in the review by Moon. The authors feel that this is enough new literature to include a section that includes as little work from that review as possible.

Line 210-212: the sentence “There are…. of the MIP.” is confusing. Please correct or rewrite it.

The sentence on line 236-237 has been altered for clarity to:

“The most common way to improve an eMIP sensor performance is through modification of the electrode.

Line 212: The sentence is started with the work from Liu et al. [79]. Immediately,  a new reference is mentiond to give information about carbazole based monomer… [80]. The next sentence is started with “This polymer…”. This polymer refers to which reference? Please consider that through the manuscript to write fluently and with a clear structure and not like a separated puzzle. 

The authors have removed this section as it did not fit with the theme of section 2.1 on neurotransmitters.

Lines 435-436: For “Electropolymerization techniques possess properties that can directly address each of these issues, allowing high affinity binding sites to be generated.” What are the reasons to prove that? References? 

The authors have added the following to improve clarity at lines 467-469:

“Electropolymerization techniques possess properties that can directly address each of these issues, by control of layer thickness [7] and the direct doping of bacteria into the polymer matrix, leading to high affinity binding sites to be generated [167].

Including the following references:

[7] Sharma, P.S.; Pietrzyk-Le, A.; D’Souza, F.; Kutner, W. Electrochemically synthesized polymers in molecular imprinting for chemical sensing. Anal. Bioanal. Chem. 2012, 402, 3177–3204, doi:10.1007/s00216-011-5696-6.

[167] Iskierko, Z.; Sharma, P.S.; Bartold, K.; Pietrzyk-Le, A.; Noworyta, K.; Kutner, W. Molecularly imprinted polymers for separating and sensing of macromolecular compounds and microorganisms. Biotechnol. Adv. 2016, 34, 30–46, doi:10.1016/J.BIOTECHADV.2015.12.002.

Line 436: What is the charged nature of the cell`s surfaces? Please add more information with references.

This has been altered in the text at lines 469-470 to:

“The negatively charged nature of the bacterial cell walls [168] helps to promote the polymerization around them [169].

Including these references:

[168]     Dickson, J.S.; Koohmaraie, M. Cell surface charge characteristics and their relationship to bacterial

attachment to meat surfaces. Appl. Environ. Microbiol. 1989, 55, 832–836.

[169]      NAKADOI, Y.; SHIIGI, H.; FURUTA, M.; TOKONAMI, S.; NAGAOKA, T.; SAIMATSU, K. Vertical Immobilization of Viable Bacilliform Bacteria into Polypyrrole Films. Anal. Sci. 2012, 28, 319, doi:10.2116/analsci.28.319.

Round  2

Reviewer 2 Report

There are still minor inconsistencies which should be eliminated:

-In this technique the sweeping of a set range of potentials, via cyclic voltammetry, between the limits of the monomer  oxidation and the reduction of the polymerized conducting polymer occurs.

Not only conducting polymers are formed, e.g. from oPD, phenol, scopoletin.

-The sensing mechanism of eMIPs commonly occurs through the same steps as seen in Figure 2.

Evaluation of the diffusional permeability is only 0NE  (indirect!) variant  of electrochemical readout, indication of electroactive products of enzymes (laccase, tyrosinase, AChE) and direct electron transfer of redox proteins e.g. cyt.c, HTHP, are DIRECT measurements.

Thus the redox markers are frequently used but not COMMONLY used

Tab.5, Ref. 174  Protein A is not a virus but a protein. It should be shifted into Tab.4

Author Response

Reviewer Comments:

1.    In this technique the sweeping of a set range of potentials, via cyclic voltammetry, between the limits of the monomer  oxidation and the reduction of the polymerized conducting polymer occurs.

Not only conducting polymers are formed, e.g. from oPD, phenol, scopoletin.

The authors agree with this statement and have removed the word conducting, the sentence between lines 176-178 no reads:

“In this technique the sweeping of a set range of potentials, via cyclic voltammetry, between the limits of the monomer oxidation and the reduction of the polymerized polymer occurs.”

2.    The sensing mechanism of eMIPs commonly occurs through the same steps as seen in Figure 2.

Evaluation of the diffusional permeability is only 0NE  (indirect!) variant  of electrochemical readout, indication of electroactive products of enzymes (laccase, tyrosinase, AChE) and direct electron transfer of redox proteins e.g. cyt.c, HTHP, are DIRECT measurements.

Thus the redox markers are frequently used but not COMMONLY used

The authors agree that there are various sensing techniques that can be used in conjunction with eMIPs. As such we have amended the sentence between lines 186-187 to improve the clarity of the work:

“One indirect sensing mechanism of eMIPs that is frequently used occurs through the same steps as seen in Figure 2.” 

3.    Tab.5, Ref. 174  Protein A is not a virus but a protein. It should be shifted into Tab.4

The authors agree that this is a protein and not a bacteria. In the paper by Khan et al. the authors directly discuss detecting the bacteria S. aureus by using an eMIP for Protein A due to its presence in the cell wall. We have added (S. Aureus) to the table to provide more clarity and this is explained in the text that relates to the paper.